

# Transposon mutagenesis reveals *Pseudomonas cannabina* pv. *alisalensis* optimizes its virulence factors for pathogenicity on different hosts

Nanami Sakata, Takako Ishiga, Haruka Saito, Viet Tru Nguyen and Yasuhiro Ishiga

Faculty of Life and Environmental Sciences, University of Tsukuba, Tsukuba, Japan

## ABSTRACT

*Pseudomonas cannabina* pv. *alisalensis* (*Pcal*), which causes bacterial blight disease of Brassicaceae, is an economically important pathogen worldwide. To identify *Pcal* genes involved in pathogenesis, we conducted a screen for 1,040 individual *Pcal* KB211 Tn*5* mutants with reduced virulence on cabbage plants using a dip-inoculation method. We isolated 53 reduced virulence mutants and identified several potential virulence factors involved in *Pcal* virulence mechanisms such as the type III secretion system, membrane transporters, transcription factors, and amino acid metabolism. Importantly, *Pcal* is pathogenic on a range of monocotyledonous and dicotyledonous plants. Therefore, we also carried out the inoculation test on oat plants, which are cultivated after cabbage cultivation as green manure crops. Interestingly among the 53 mutants, 31 mutants also exhibited reduced virulence on oat seedlings, indicating that *Pcal* optimizes its virulence factors for pathogenicity on different host plants. Our results highlight the importance of revealing the virulence factors for each plant host-bacterial interaction, and will provide new insights into *Pcal* virulence mechanisms.

## INTRODUCTION

Gram-negative *Pseudomonas syringae* phytopathogenic bacteria are known to cause diseases in a wide range of economically important host plants with a variety of diseases symptoms, including blight, cankers, leaf spots, and galls. Currently, approximately 60 pathovars (pv) have been identified, and each has developed its own infection strategies to adapt to each host plant (*Xin, Kvitko & He, 2018*). In a successful disease cycle, *P. syringae* has two interconnected lifestyles: the epiphytic phase upon on the surface of plant tissues, and the endophytic phase in the intercellular apoplastic space (*Xin & He, 2013*; *Xin, Kvitko & He, 2018*). The plant surface is generally considered to be suboptimal for microbes, and successful epiphytes have to evolve mechanisms that allow them to survive in severe conditions (*Lindow & Brandl, 2003*). After sufficient epiphytic proliferation, bacteria enter plant leaves through natural openings, mainly stomata, multiply in the intercellular space,

Corresponding author
Yasuhiro Ishiga,
ishiga.yasuhiro.km@u.tsukuba.ac.jp

and eventually produce chlorotic halos and necrotic lesions (*Hirano & Upper, 2000*). However, the virulence factors involved in these processes largely remain unclear.

To identify pathogenicity related genes, several forward screens have been conducted in the past decade. Generation and characterization of Tn*5* insertion mutation libraries is one of the most efficient screens. For example, 2,000 Tn*5* mutations were generated in *Pseudomonas syringae* pv. *phaseolicola* and inoculated into bean pods (*Phaseolus vulgaris* cv. Kinghorn wax). Four strains showed no pathogenicity in host plants, and no hypersensitive response in nonhost plants (*Somlyai et al., 1986*). Tn5-induced insertion mutations were also generated in *Pseudomonas syringae* pv. *tomato*. In that study, 920 mutated strains were inoculated onto tomato plants (*Lycopersicon esculentem* Mill. 'Bonny Best'). Nine strains showed reduced virulence, and 5 out of 9 strains also showed no hypersensitive response in nonhost plants (*Cuppels, 1986*). Screening of 947 Tn*5* mutants by dip-inoculation onto *Arabidopsis thaliana* was also conducted, and 37 strains were identified as reduced virulence mutants (*Brooks et al., 2004*). The study also indicated that dip-inoculation, which is similar to natural conditions, allows for the isolation of mutants that are involved in the whole infection process, including epiphytic growth, entry, and colonization in the apoplast (*Brooks et al., 2004*). A high-throughput forward genetic screen identified genes required for the virulence of *Pseudomonas syringae* pv. *maculicola* ES4326 (*Pseudomonas cannabina* pv. *alisalensis* ES4326) on *Arabidopsis* (*Schreiber et al., 2012*). In that study, approximately 12,600 transposon disruptants were screened in the liquid assay, and 40 mutants with reduced virulence were determined. It is noteworthy that forward genetic screens have provided valuable insights into the various virulence factors required for the virulence of bacterial pathogens.

Disease outbreaks caused by new *P. syringae* isolates are a problem worldwide. One of them is the devastating kiwifruit canker outbreak in New Zealand and Europe, which is caused by *Pseudomonas syringae* pv. *actinidiae* (*Mazzaglia et al., 2012*; *Butler et al., 2013*; *McCann et al., 2013*). Another example is bacterial leaf spot and blight of Brassicaceae caused by *Pseudomonas syringae* pv. *maculicola* (*Psm*) and *Pseudomonas cannabina* pv. *alisalensis* (*Pcal*). These diseases are very important and cause significant damage to crucifer crops worldwide (*Sarris et al., 2013*; *Takahashi et al., 2013*; *Takikawa & Takahashi, 2014*). In 2002, *Cintas, Koike & Bull (2002)* studied blight leaf spot of broccoli (*B. orelacea* var. *italica*) and broccoli raab (*B. rapa* subsp. *rapa*) in California, and proposed a new name *P. syringae* pv. *alisalensis* for the pathogen, which is distinguishable from *Psm* (*Cintas, Koike & Bull, 2002*). Subsequently, *P. syringae* pv. *alisalensis* was reclassified as *Pseudomonas cannabina* pv. *alisalensis* (*Pcal*) (*Bull et al., 2010*). *Pcal* has similar characteristics with *Psm*, but is distinct in some bacteriological characteristics, genetic traits, and its ability to infect gramineous plants such as oat (*Avena sativa*) and timothy (*Phleum pretense*) (*Cintas, Koike & Bull, 2002*; *Bull et al., 2010*). In addition, *Pcal* tends to cause more severe disease symptoms compared to *Psm* (*Takikawa & Takahashi, 2014*). Recent work demonstrated that some isolates that had been identified as *Psm* are actually *Pcal* (*Takikawa & Takahashi, 2014*). In Nagano, Japan, an outbreak of bacterial blight occurred and the isolates were identified as *Pcal* (*Ishiyama et al., 2013*). *Pcal* can infect not only a wide range of Brassicaceae but also green manure crops grown naturally before and after

**Table 1** Bacterial strains and plasmid used in this study.

| Bacterial strain or plasmid | Relevant characteristics | Reference or source |
|---|---|---|
| *E. coli* strain | | |
| DH5α | F⁻λ⁻φ80dLacZΔM15 Δ(lacZYA-argF)U169 recA1 endA1 hsdR17 (rK⁻mK⁺) supE44 thi-1gyrA relA1 | Takara, Kyoto, Japan |
| S17-1 | Thi pro hsdR⁻hsdM⁺ recA [chr::RP4-2-Tc::Mu-Km::Tn7] | *Schäfer et al. (1994)* |
| *P. cannabina* pv. *alisalensis* | | |
| Isolate KB211 | Wild type, Rif^r | Nagano vegetable and ornamental crops expe riment station |
| KB211-VR series | Whole genome Tn5 transposon library, Rif^r, Km^r, Cm^r | This work |
| Plasmid | | |
| pBSLC1 | Transposon vector constructed by ligation of pBSL118 and pHSG396 at EcoR I site, Amp^r, Km^r, Cm^r | *Sawada et al. (2018)* |

**Notes.**
Amp^r ampicillin resistance, Cm^r chloramphenicol resistance, Km^r kanamycin resistance, Rif^r rifampicin resistance.

crucifer crops, and causes brown spot on oat plants (*Ishiyama et al., 2013*). Because of this unique characteristic, *P. cannabina* has potential to be a model for studying plant-bacterial interactions across divergent host plants, and can serve as a parallel model system to *P. syringae* for understanding virulence and host range evolution (*Sarris et al., 2013*).

Chemical treatments such as copper fungicides and antibiotics are popular strategies for bacterial disease control. However, a *Pcal* strain resistant against these chemicals has already appeared (*Takahashi et al., 2013*). Therefore, we need to develop more efficient and sustainable strategies for bacterial disease control. Revealing the infection mechanism is an important component of developing new strategies for disease control. In spite of this, *Pcal* virulence factors and mechanisms have not been well studied and remain unclear. In order to identify genes involved in *Pcal* pathogenicity, we established a system for screening *Pcal* KB211 Tn5 insertion mutants with reduced virulence on cabbage. Here, we provide several potential virulence factors involved in *Pcal* virulence mechanisms, and the basis of the adaptation of virulence mechanisms to different hosts.

## MATERIALS AND METHODS

### Bacterial strains, plasmids, and growth conditions

The bacterial strains and plasmids used in this study are described in Table 1. *Pseudomonas cannabina* pv. *alisarensis* (*Pcal*) strain KB211 was used as the pathogenic strain to inoculate cabbage and oat. *Pcal* strains were grown on mannitol-glutamate (MG; *Keane, Kerr & New, 1970*) medium, Luria-Bertani (LB; *Sambrook, Fritsch & Maniatis, 1989*) medium, or King's B (KB; *King, Ward & Raney, 1954*) medium at 28 °C, and *E. coli* strains were grown on LB medium at 37 °C. Antibiotics used for selection of *Pcal* strains and *E. coli* strains included (in μl/ml): rifampicin, 50; kanamycin, 50; ampicillin, 50; chloramphenicol, 10. Before *Pcal* inoculation, bacteria were suspended in sterile distilled $H_2O$, and the bacterial cell densities at 600 nm ($OD_{600}$) were measured using a JASCO V-730 spectrophotometer (JASCO, Tokyo, Japan).

### Generation of a *Pcal* genomic Tn*5* mutant library

The transposon was introduced into *Pcal* KB211 by conjugation with *E. coli* S17-1 which possessed pBSLC1 (*Sawada et al., 2018*), and the insertion region was integrated into the *Pcal* chromosome randomly (Fig. 1). Replica plates for all transconjugants were made and used for the inoculation assay.

### Plant materials

Cabbage (*Brassica oleracea* var. *capitate*) cv. Kinkei 201 and oat (*Avena strigosa*) cv. Hayoat plants were used for *Pcal* virulence assays. Tobacco (*Nicotiana tabacum*), cv. Xanthi plants were used for the *Pcal* hypersensitive reaction (HR) cell death assay. Cabbage, oat, and tobacco plants were grown from seed at 24 °C with a light intensity of 200 $\mu Em^{-2}s^{-1}$ anda 16 h light/8 h dark photoperiod. Cabbage and oat seedlings were used for inoculation assays around two weeks after germination. Tobacco plants were used for the HR cell death assay around four weeks after germination.

### Growth curve assay

*Pcal* strains including the wild-type and mutants were grown at 28 °C for 24 h in KB broth. The strain suspensions were adjusted to an $OD_{600}$ of 0.01 with fresh KB broth, and the bacterial growth dynamics were measured at $OD_{600}$ for 12 h, 24 h, and 48 h.

### Bacterial inoculation methods

To assay for disease on cabbage and oat seedlings, dip-inoculation was conducted by soaking plants in bacterial suspensions ($5 \times 10^7$ CFU/ml) containing 0.025% Silwet L-77 (OSI Specialities, Danbury, CT, USA). The plants were then incubated in growth chambers at ~100% RH for the first 24 h, then at ~70% RH for the rest of the experimental period. The disease symptoms on cabbage seedlings were evaluated 5 days post-inoculation. The symptoms on cabbage were classified into 4 stages based on disease severity; 0, 1, 2, and 3 (Fig. 1). For screening of reduced virulence mutants on cabbage seedlings, the inoculation assay was repeated more than three times, and mutants showing an average score of less than 2 were selected as reduced virulence strains. The disease symptoms on oat seedlings were evaluated 4 days post-inoculation. The symptoms on oat were classified into 3 stages based on disease severity; 0, 1, and 2 (Fig. 1). For screening of reduced virulence mutants on oat seedlings, the inoculation assay was repeated at least three times, and mutants showing an average score of less than 1.2 were selected as reduced virulence strains.

To assess bacterial growth in cabbage and oat seedlings, the internal bacterial population was measured after dip-inoculation. Inoculated plants were collected and the total of two inoculated leaves were measured 5 days and 4 days after inoculation on cabbage and oat, respectively. The leaves were surface-sterilized with 10% $H_2O_2$ for 3 min. After washing three times with sterile distilled water, the leaves were homogenized in sterile distilled water, and diluted samples were plated onto solid KB agar medium. Two or three days after plating of the diluted samples, the bacterial colony forming units (CFUs) were counted and normalized as CFU per gram, using the total weight of the leaves. The bacterial populations were evaluated in three independent experiments.

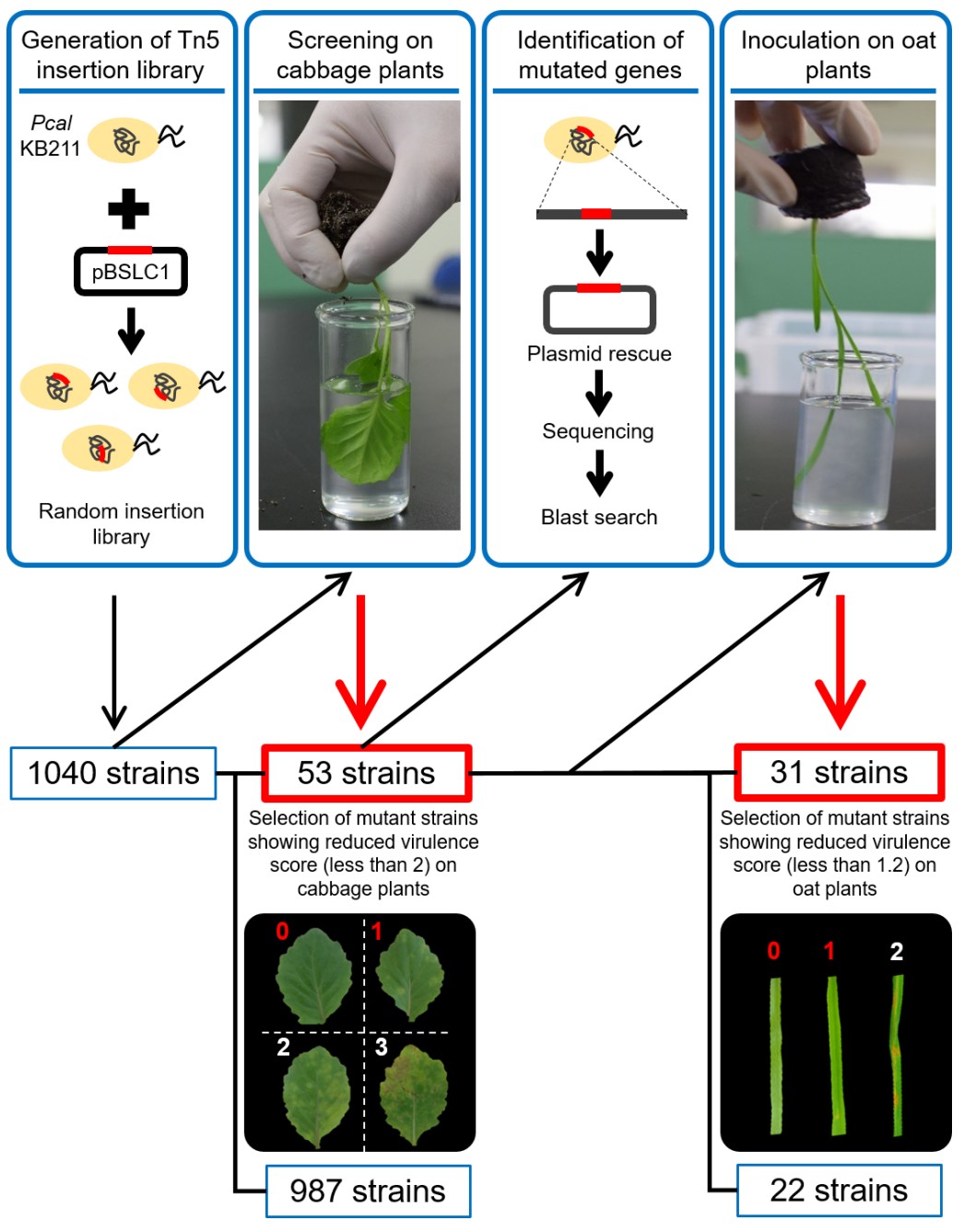

**Figure 1** **The workflow of this study.** The transposon was introduced into *Pcal* KB211 by conjugation with *E. coli* S17-1 which possessed pBSLC1, and the insertion region was integrated into the *Pcal* chromosome randomly. To assay for disease on cabbage and oat seedlings, 1,040 individual *Pcal* KB211 Tn5 mutants were used to dip-inoculate cabbage plants, which were about 16 days old. Then, disease symptoms were observed five days post-inoculation (dpi). Mutant strains that caused little or no chlorosis (virulence score less than 2) were chosen for further analyses. A total of 53 mutants showed reduced virulence on cabbage and the mutated genes were determined. To identify the mutated (continued on next page...)

**Figure 1 (…continued)**
genes, the resultant DNA was ligated with T4 DNA ligase, then introduced into *E. coli* DH5α competent cells. A *Pseudomonas* Genome DB BLAST search (http://www.pseudomonas.com/blast/setnblast) was utilized to identify the function of the mutated genes. We also conducted dip-inoculation of oat plants, which were about 12 days old, with these 53 mutants. Disease symptoms were evaluated four days post-inoculation. 31 mutant strains also showed reduced virulence on oat plants, which caused little or no brown spots (virulence score less than 1.2). Inoculation assay was repeated more than three times and average score was calculated.

To analyze *Pcal*-induced HR cell death in tobacco leaves, bacterial suspensions ($5 \times 10^7$ CFU/ml) of *Pcal* wild-type and virulence mutants were prepared and infiltrated into leaves using a one ml needleless syringe. HR cell death was observed 24 h after infiltration.

## Plasmid rescue of transposon-integrated regions and sequencing analysis to identify insertion sites

Genomic DNA of the mutants which showed reduced virulence on cabbage were purified by using a Nucleospin Microbial DNA Kit (TaKaRa, Ohtsu, Japan) and digested with Hind III, Xh I, Sph I, Kpn I, Sal I, Xba I, or Hinc II (TaKaRa). The resultant DNA was ligated with T4 DNA ligase (Ligation-convenience kit, Nippon Gene, Tokyo, Japan), then introduced into *E. coli* DH5α competent cells. Plasmid DNA was purified from the transformants, and transposon-insertion sites were identified by sequencing with the M13 forward primer. A *Pseudomonas* Genome DB BLAST search (http://www.pseudomonas.com/blast/setnblast) was utilized to identify the function of the mutated genes (Fig. 1).

# RESULTS

## Isolation of *P. cannabina* pv. *alisalensis* KB211 mutants with reduced virulence

We conducted a screen for mutants with reduced virulence to identify *Pcal* KB211 pathogenesis related genes. To this point, 1,040 individual *Pcal* KB211 Tn*5* mutants were dip-inoculated onto cabbage plants, and we examined for disease symptoms 5 days after inoculation. Disease symptoms on cabbage caused by wild-type *Pcal* KB211 included chlorosis and necrosis (Fig. 1). A total of 53 mutants showed reduced virulence in multiple infection experiments. These Tn*5* mutants were also tested for their ability to grow on MG minimal medium. Four mutants were determined to be partial auxotrophs because they could not grow on MG medium (Table 2).

## Identification of genes disrupted by Tn*5* insertions

Next, we identified the Tn*5* insertion sites of 53 mutant strains which showed reduced virulence. Our results suggest that various virulence factors are needed for the *Pcal* infection process (Table 2). We recognized several genes which were previously identified with known function, and novel genes which were identified with unknown function. Known virulence factors included genes involved in the type III secretion system (T3SS), the type IV secretion system (T4SS), and multidrug resistance efflux pumps. Several genes were identified as a hypothetical protein, whose function was unknown. Furthermore, we identified regulators including transcription factors and small signal molecules essential

Sakata et al. (2019), PeerJ, DOI 10.7717/peerj.7698

**Table 2 Characteristics of selected *Pcal* KB211 transposon disruption mutants (*N* = 53).** The mutated genes were determined by using *Pseudomonas* Genome DB BLAST search based on the genome database of *Pseudomonas syringae* pv. *maculicola* ES4326 (*Pseudomonas cannabina* pv. *alisalensis* ES4326). Functional category annotations for *Pcal* KB211 genes are primary based on COG (*Chen et al., 2019*) and KEGG (*Kanehisa & Goto, 2000*) annotations. Asterisks represent the mutants which virulence score was less than 1.2 on oat. The Tn*5* mutants were also tested for their ability to grow on MG minimal medium. + represents positive growth. − represents negative growth.

| Classification | Mutant | Locus | Description | Virulence score | | Growth in MG |
|---|---|---|---|---|---|---|
| | | | | Cabbage | Oat | |
| Amino acid metabolism and transport | NF34 | PMA4326_25462 | 3-phosphoglycerate dehydrogenase | 0 | 0.29* | − |
| | NN31 | PMA4326_01907 | D-amino acid dehydrogenase small subunit | 0.29 | 0* | + |
| | NI13 | PMA4326_20937 | N-acetyl-gamma-glutamyl-phosphate reductase | 0 | 0.14* | − |
| | NF2 | PMA4326_02177 | Tryptophan synthase subunit alpha | 0 | 0.14* | − |
| | NH11 | PMA4326_18563 | N-ethylammeline chlorohydrolase | 1 | 2 | + |
| | NM37 | PMA4326_04881 | Taurine catabolism dioxygenase TauD | 0.75 | 1.67 | + |
| DNA processing and modification | NN6 | PMA4326_27962 | Error-prone DNA polymerase | 0.43 | 0.43* | + |
| | NV6 | PMA4326_22604 | Helicase | 1.62 | 1.5 | + |
| | NE29 | PMA4326_20877 | Helicase UvrD | 0.33 | 2 | + |
| | NI5 | PMA4326_29937 | Integrase | 1.2 | 2 | + |
| | NC22 | PMA4326_30012 | Mobilization protein | 1.33 | 2 | + |
| | NN12 | PMA4326_19710 | N-methyltransferase | 1.29 | 1.5 | + |
| Transcriptional regulator | NL8 | PMA4326_16681 | ArsR transcriptional factor | 0.57 | 0.71* | + |
| | NV13 | PMA4326_26857 | Fis family transcriptional factor | 1 | 0.86* | + |
| | NK1 | PMA4326_02287 | HexR transcriptional factor | 0.43 | 0.29* | + |
| | NN14 | PMA4326_08221 | LysR family transcriptional factor | 0.71 | 0.57* | + |
| | NN5 | PMA4326_21784 | AraC family transcriptional regulator | 1 | 2 | + |
| Type IV secretion system | NK16 | PMA4326_28915 | DNA topoisomerase III | 0.29 | 0.58* | + |
| | NM29 | PMA4326_28915 | DNA topoisomerase III | 1 | 1* | + |
| | NA13 | PMA4326_28915 | DNA topoisomerase III | 1.33 | 1.5 | + |
| | NA4 | PMA4326_28915 | DNA topoisomerase III | 0.67 | 2 | + |
| | NV27 | PMA4326_28955 | Relaxase | 0.63 | 1.14* | + |
| Nucleotide metabolism and transport | NL37 | PMA4326_26032 | D-tyrosyl-tRNA(Tyr) deacylase | 1 | 0.57* | + |
| | NF22 | PMA4326_22906 | Phosphoribosylaminoimidazole carboxylase ATPase subunit | 0.17 | 0.43* | − |
| | NL3 | PMA4326_05636 | Polynucleotide phosphorylase/polyadenylase | 0 | 0.13* | + |
| | NH26 | PMA4326_04134 | Dehydrogenase | 1 | 2 | + |
| | NM30 | PMA4326_19765 | Phosphoribosyltransferase | 1.25 | 2 | + |
| Lipid metabolism and transport | NM2 | PMA4326_20637 | Acetyltransferase family protein | 0.71 | 0.29* | + |
| | NN3 | PMA4326_17521 | Lipid kinase | 0.57 | 0.43* | + |
| | NM18 | PMA4326_10330 | Acyltransferase | 1 | 1.67 | + |

**Table 2** (*continued*)

| Classification | Mutant | Locus | Description | Virulence score | | Growth in MG |
|---|---|---|---|---|---|---|
| | | | | **Cabbage** | **Oat** | |
| Carbohydrate metabolism and transport | NM10 | PMA4326_26042 | Glycogen phosphorylase | 0.57 | 1* | + |
| | NN1 | PMA4326_09615 | Glycerol kinase | 1.5 | 1.67 | + |
| Energy generation | NU21 | PMA4326_20235 | Malate:quinone oxidoreductase | 0.38 | 0.29* | + |
| | NV7 | PMA4326_01045 | Polyphosphate kinase | 0.83 | 0.86* | + |
| Membrane transport | NU19 | PMA4326_12408 | RND transporter | 0.75 | 1* | + |
| | NU37 | PMA4326_26242 | Sulfonate ABC transporter substrate-binding protein | 0.25 | 1.14* | + |
| Peptidoglycan/cell wall polymers | NC15 | PMA4326_15504 | D-alanyl-D-alanine carboxypeptidase | 0 | 1.14* | + |
| | NN13 | PMA4326_23061 | N-acetylmuramoyl-L-alanine amidase | 0.57 | 1.14* | + |
| Signal transduction mechanisms | NM38 | PMA4326_16771 | Methyl-accepting chemotaxis protein | 0.57 | 0.57* | + |
| | NH7 | PMA4326_04474 | HDOD domain protein | 0.33 | 2 | + |
| Type III secretion system | NB35 | PMA4326_02782 | Type III secretion protein HrcQb | 0 | 0.5* | + |
| Cofactor metabolism | NC12 | PMA4326_20982 | Lipoate-protein ligase | 0.83 | 1.14* | + |
| Nitrogen metabolism | NU13 | PMA4326_16286 | Nitrate reductase | 0.5 | 1.14* | + |
| Stress resistance | NU25 | PMA4326_00370 | 16S rRNA methyltransferase | 0.13 | 0.71* | + |
| Iron metabolism and transport | NI6 | PMA4326_03299 | Heme oxygenase | 1.4 | 2 | + |
| Hypothetical | NN20 | PMA4326_19408 | Hypothetical protein | 0.29 | 0.43* | + |
| | NV25 | PMA4326_10750 | YD repeat-containing protein | 1 | 0.57* | + |
| | NI7 | PMA4326_30377 | Hypothetical protein | 1.63 | 1.25 | + |
| | NL30 | PMA4326_27167 | Hypothetical protein | 1.25 | 1.67 | + |
| | NM27 | PMA4326_29225 | Hypothetical protein | 0.25 | 1.67 | + |
| | NM32 | PMA4326_27567 | Hypothetical protein | 1 | 2 | + |
| | NV33 | PMA4326_08475 | Hypothetical protein | 1.2 | 2 | + |
| | NK39 | PMA4326_18613 | Rhodanese-like domain protein | 0 | 1.67 | + |

for the expression of genes required for survival during environmental stress conditions. Other Tn5 insertions included genes related to primary metabolism such as amino acid metabolism, lipid metabolism, carbohydrate metabolism, nucleotide metabolism, and energy generation (Table 2).

Four partial auxotroph mutants had Tn5 insertions in genes encoding tryptophan synthase subunit alpha, phosphoribosylaminoimidazole carboxylase, 3-phosphoglycerate dehydrogenase, and N-acetyl-gamma-glutamyl-phosphate reductase. The mutants which showed complete pathogenicity impairment on cabbage (virulence score 0) had Tn5 insertions in genes encoding T3SS proteins HrcQb, D-alanyl-D-alanine carboxypeptidase, tryptophan synthase subunit alpha, 3-phosphoglycerate dehydrogenase, N-acetyl-gamma-glutamyl-phosphate reductase, and polynucleotide phosphorylase/polyadenylase. The category of "amino acid metabolism and transport" were enriched in these genes (Table 2).

### Virulence on oat seedlings

Pcal KB211 is known to infect oat plants (Ishiyama et al., 2013), so we constructed a virulence test on oat seedlings for these 53 mutants as well as wild-type Pcal KB211. Disease symptoms on oat seedlings caused by wild-type Pcal KB211 showed brown spot. Among 53 mutants, 31 mutants also exhibited reduced virulence on oat seedlings in multiple infection experiments (Table 2; Figs. S1 and S2). We focused our analysis on genes contributing to successful infection on both cabbage and oat seedlings. We also investigated the bacterial growth of wild-type Pcal and these 31 mutant strains in KB liquid medium. Most of the mutants grew similarly to wild-type Pcal at 48 h, except for NL8 and NN13 (Fig. S3).

Genes required for the biosynthesis of several different amino acids were important for infection on both cabbage and oat seedlings (Table 2). Four mutants impaired in amino acid synthesis showed a significant reduction in disease symptom production on both plants, and three of the four were partial auxotrophs. Genes required for regulation were also important for infection on both host plants, and four transcriptional factors were identified: HexR, ArsR, LysR family, and the Fis family. Additionally, genes encoding the T3SS were also highly important for infection on both cabbage and oat. To confirm this, mutants potentially defective in the T3SS were assayed for their ability to elicit macroscopic tissue collapse indicative of HR on nonhost tobacco. The mutants failed to cause HR as we expected (Fig. S4). Furthermore, genes encoding ABC transporters and RND transporters were included in the common virulence factors. We also found that genes encoding polyphosphate kinase belonged to this group.

### Bacterial growth of the virulence mutants in plant tissue

To determine whether the reduction in disease symptom production was correlated with reduced multiplication in plant tissue, 31 mutants were analyzed for their ability to grow in cabbage and oat seedlings after dip-inoculation. The wild-type Pcal KB211 multiplied to 9.82 (Log [CFU/g]), and all the mutants except NL37 and NN13 exhibited a significant reduction in growth relative to wild-type Pcal KB211 on cabbage leaves (Fig. 2A). Although there was no significant differences in bacterial growth between these mutants and wild-type Pcal KB211, decreasing trends were observed. In addition, the wild-type Pcal

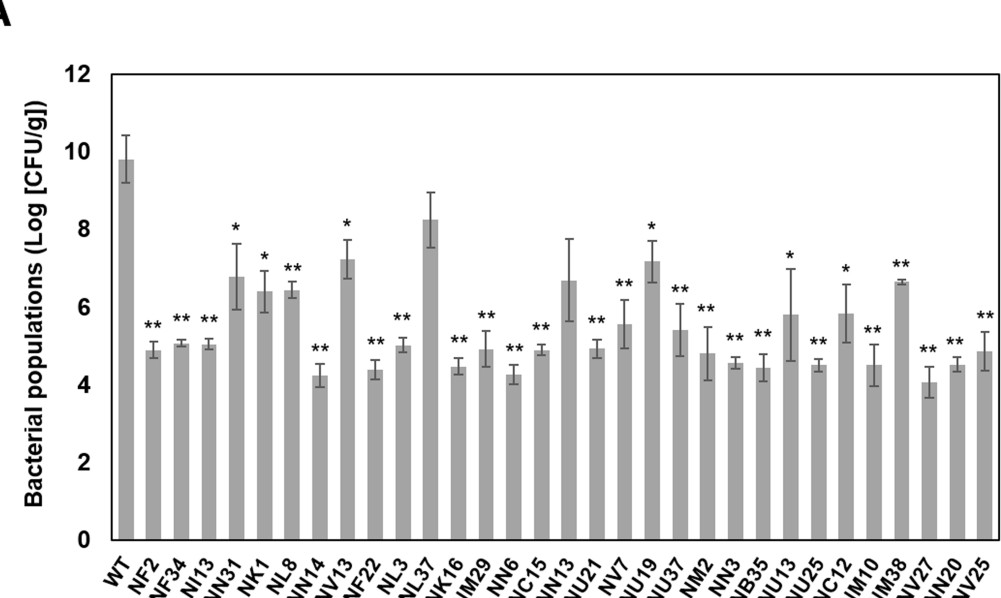

**A**

**B**

**Figure 2** **Bacterial population dynamics in cabbage (A) and in oat (B) inoculated with bacterial suspensions of wild-type *Pcal* KB211 and mutants containing 0.025% Silwet L-77 at a concentration of $5 \times 10^7$ CFU/ml.** The bacterial populations were obtained by homogenizing the inoculated leaves after surface-sterilization and plating dilutions to selective media at 5 days post-inoculation (dpi) (cabbage) and at 4 dpi (oat). Vertical bars indicate the standard error for three independent experiments. Asterisks indicate a significant difference from the WT and each mutant in a *t* test (* < 0.05, ** < 0.01).

KB211 multiplied to 8.44 (Log [CFU/g]) and all the mutants except NC15, NU37, NU13, and NV27 exhibited a significant reduction in growth relative to wild-type *Pcal* KB211 on oat leaves (Fig. 2B). The wild-type *Pcal* bacterial populations on cabbage leaves were greater than that on oat leaves. Bacterial populations of NL37, encoding D-tyrosyl-tRNA (Tyr) deacylase, on cabbage plants were about $10^2$-fold greater than that on oat plants. In contrast, bacterial populations of NC15, encoding D-alanyl-D-alanine carboxypeptidase, on cabbage plants were about $10^3$-fold less than that on oat plants.

## DISCUSSION

Although *Pcal* is one of the most economically important bacterial pathogens worldwide, genetic studies have not been performed to understand its virulence mechanisms. To identify *Pcal* genes that play an important role in pathogenesis, we isolated *Pcal* KB211 mutants which exhibited reduced virulence on cabbage and identified several potential virulence factors (Table 2; Fig. 3). Although forward genetic screens are generally laborious, these screens have provided valuable insights into the various factors required for the virulence of bacterial phytopathogens. *Schreiber et al. (2012)* utilized a high-throughput, liquid media-based assay to screen *Psm* ES4326 (*Pcal* ES4326) transposon mutants to identify genes required for virulence on *A. thaliana*. Genes identified through this screen were involved in the T3SS, periplasmic glucan biosynthesis, flagellar motility, and amino acid biosynthesis. They demonstrated that the ability of *Psm* ES4326 to synthesize specific amino acids strongly influences its proliferation in *Arabidopsis* leaf tissues. They also reported that nearly half of these mutant strains involved genes associated with the T3SS (*Schreiber et al., 2012*). In addition, *Brooks et al. (2004)* conducted a screen for *Pst* DC3000 Tn*5* mutants by dip-inoculation on *A. thaliana*. In this study, 947 individual *Pst* DC3000 Tn*5* mutants of were dip-inoculated onto *A. thaliana*, and 37 mutants exhibited reduced virulence. They found that mutants disrupted genes involved in the T3SS, the phytotoxin coronatine, and amino acid biosynthesis, and around 15% of these genes were related to the T3SS and coronatine biosynthesis respectively (*Brooks et al., 2004*). Similarly, we also identified genes involved in the T3SS, periplasmic glucan biosynthesis, and amino acid biosynthesis in our screen (Table 2; Fig. 3). However, our results showed great diversity of virulence factors, and provided novel insights into the factors not previously known to be associated with *Pseudomonas* virulence. Previous studies identified many genes involved in the T3SS, but we identified only one T3SS related mutant (Table 2). Therefore, it is important to note that these differences might be explained by the differences in the host plant species and pathovar studied. Together, these results suggest the importance of exploring the virulence factors of each bacterial pathogen.

Interestingly, our results revealed great diversity in *Pcal* KB211 potential virulence factors (Table 2; Fig. 3). Furthermore, our results revealed that nearly half of the genes making contributions to virulence in cabbage also make virulence contributions in oat. We also observed that wild-type *Pcal* bacterial populations in cabbage plants were greater than those in oat plants (Fig. 2). *Pcal* is originally a bacterial pathogen which causes diseases on cruciferous plants (*Ishiyama et al., 2013*). Taken together, our results implied the

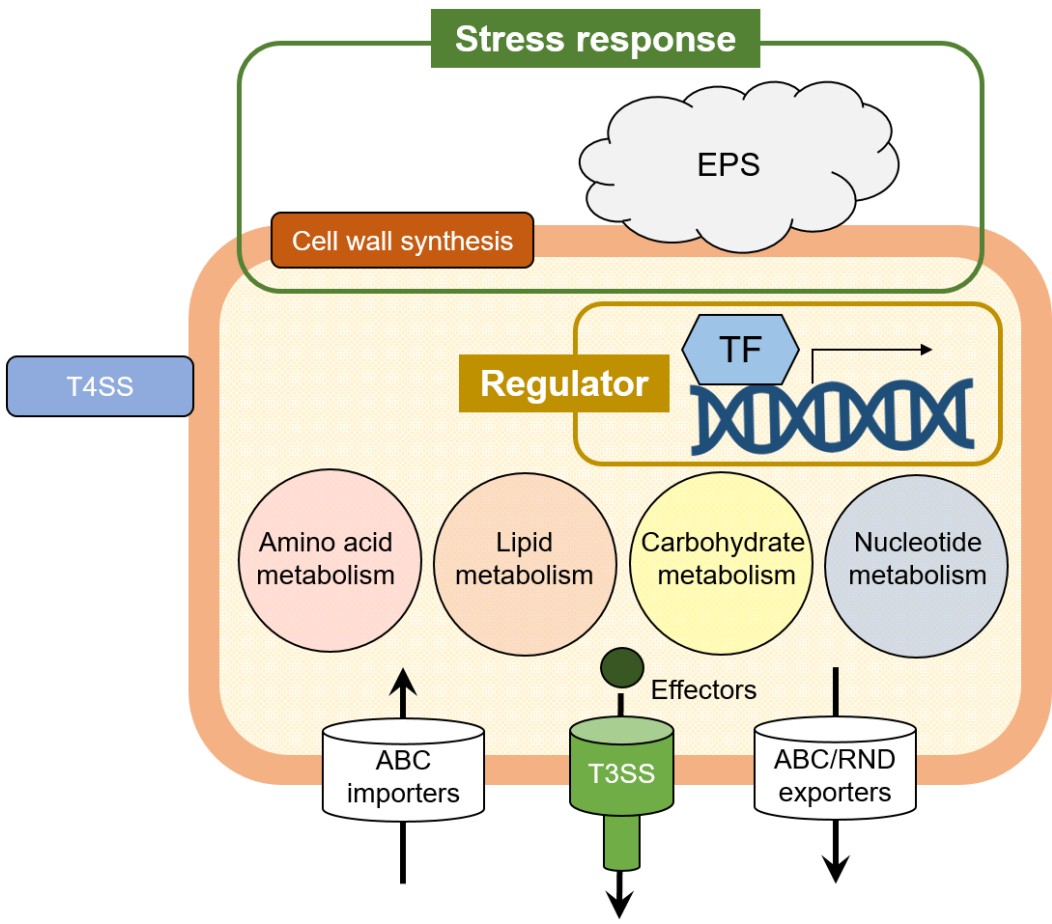

**Figure 3** **The illustrated *Pcal* virulence factors based on this study.** Cell wall synthesis and exopolysaccharide (EPS) genes are involved in stress response. Several transcriptional factors (TFs) regulate the *Pcal* KB211 virulence mechanisms. Primary metabolism, including amino acid metabolism, lipid metabolism, carbohydrate metabolism, and nucleotide metabolism also play an important role in pathogenicity. *Pcal* KB211 contains ABC (ATP binding cassette) transporters and RND (resistance-nodulation-cell division) transporters. The T3SS (Type III secretion system) and T4SS (Type IV secretion system) are involved in *Pcal* KB211 virulence mechanisms.

possibility that bacteria easily infect cabbage and have a strong infection strategy against cabbage compared to oat. Moreover, our results showed 31 mutants showing reduced virulence on both plants, also showed decreased multiplication *in planta* (Table 2; Fig. 2), suggesting that the decreased virulence could result from an overall diminished fitness. In this study, we highlight that *Pcal* optimizes its virulence factors for pathogenicity on cabbage and oat plants. Further spatiotemporal analysis for virulence mutants will allow for an enhanced understanding of *Pcal* pathogenicity.

We demonstrated that the T3SS is an essential factor for the infection process on both cabbage and oat plants (Table 2). Especially, the *hrcQb* mutant showed complete pathogenicity impairment on cabbage and a significant reduction of bacterial growth on both cabbage and oat (Table 2; Fig. 2). The T3SS is well known as a virulence factor of many

gram-negative plant and animal pathogenic bacteria (*Büttner & He, 2009*). In this study, we did not find any hypothetical proteins showing similarity to known T3SS effectors, nor HrpL binding sites in the sequencing of the insertion sites. Further characterization of the T3SS and Type III secreted effectors (T3SEs) in *Pcal* KB211 will be needed.

In this screening, we identified virulence genes which encode ABC transporters and RND transporters (Table 2). Efflux systems can directly export virulence determinants and contribute to bacterial pathogenesis in *P. aeruginosa* (*Hirakata et al., 2002*). Genes for multidrug resistance in *E. coli* are involved in biofilm formation (*Matsumura et al., 2011*). In *P. syringae*, mutant strains that are defective in genes for the multidrug efflux pump had enhanced sensitivity to a wide array of antimicrobials and decreased multiplication (*Stoitsova et al., 2008*). Importantly, both disease symptom production and bacterial growth of the mutant disrupted ABC transporter were significantly reduced on cabbage plants compared to the reduction on oat plants (Fig. 2). As ABC transporters are known to intake nutrients, this difference can be caused by differences in nutrient composition between cabbage and oat. Although the reduction level with respect to disease symptoms was different, these two transporters can play important roles in *Pcal* pathogenicity.

We also identified several genes which are not considered closely related to pathogenicity including genes for amino acid metabolism, cell wall synthesis, and transcriptional factors (Table 2; Fig. 3). Especially, the mutants which are related to amino acid metabolism showed complete pathogenesis impairment similar to the T3SS mutant, suggesting that amino acid biosynthesis has an important role in bacterial pathogenicity. Genes involved in tryptophan synthesis had a great effect on disease symptom development and bacterial population when disrupted in our study, on both cabbage and oat plants (Table 2; Fig. 2). Tryptophan is utilized in downstream biosynthetic pathways, such as the auxin pathway, and has been shown to contribute to *P. syringae* pv. *syringae* B728a epiphytic fitness (*Lindow & Brandl, 2003*; *Duong, Jensen & Stevens, 2018*). In *Pst* DC3000, pathogen-derived auxin promotes virulence by suppressing SA-mediated defenses (*McClerklin et al., 2018*). Genes within the serine biosynthetic pathway, which encode 3-phosphoglycerate dehydrogenase (SerA), also had a great effect on disease symptom development and bacterial growth (Table 2; Fig. 2). SerA is associated with bacterial motility and adherence in *P. aeruginosa* (*Yasuda et al., 2017*). Further characterization will be needed for a *Pcal serA* mutant with respect to motility, including swimming, swarming, and twitching. Interestingly, four mutants, including these two, could not grow on MG medium, and showed partial auxotrophy. It is known that the leaf surface provides limited nutrient resources for bacteria (*Lindow & Brandl, 2003*). Rico and Preston reported that *Pst* DC3000 uses amino acids that are abundant in the plant apoplast (*Rico & Preston, 2008*). Moreover, Schreiber et al. demonstrated that the ability of *Psm* ES4326 to synthesize specific amino acid strongly influences its proliferation in *Arabidopsis* leaf tissue (*Schreiber et al., 2012*). Further substantiation on the role of amino acid biosynthesis in pathogenicity will be a priority question in our future studies.

We also identified regulators including transcriptional factors and small signal molecules essential for the expression of genes required for survival during environmental stress conditions (Table 2). Our results indicated that transcription factors involved in the

oxidative stress response might be important in *Pcal* virulence. OxyR regulates gene expression related to ROS-detoxifying enzymes, including catalases, and is required for full virulence in *Pst* DC3000 (*Ishiga & Ichinose, 2016*). In addition to OxyR, the HexR transcriptional regulator regulates numerous genes under oxidative stress conditions in *P. putida* (*Kim, Jeon & Park, 2008*; *Kim & Park, 2014*). The ORFs encoding putative homologs to *P. putida* HexR are present in the *Pst* DC3000 genome (*Ishiga & Ichinose, 2016*). In our study, the mutant disrupted genes coding HexR clearly showed reduced disease symptoms as well as reduced bacterial population both in cabbage and oat plants (Table 2; Fig. 2), indicating that HexR contributes to *Pcal* virulence. It was demonstrated that HexR might be a dual-sensing regulator of *zwf-1* induction, and is able to respond to both carbohydrate metabolism and oxidative stress (*Kim, Jeon & Park, 2008*). Further investigation will be needed to understand whether HexR regulates the signal pathways involved in carbohydrate metabolism and oxidative stress. We also discovered that mutants which are impaired in LysR encoding genes showed reduced disease symptoms and bacterial population both in cabbage and oat plants (Table 2; Fig. 2). Taken together, these results suggest that the management of oxidative stress during infection by these transcription factors can be important in *Pcal* pathogenicity.

In summary, our results identified several genes assuming unimpaired growth in culture which were involved in *Pcal* virulence on cabbage, and implies the differential requirement for virulence factors depending on host plants. Moreover, predicted functions based on sequence analysis suggest that these virulence factors may be involved at different stages of the infection process. We need to conduct further screening to elucidate the whole infection mechanism of *Pcal*. In addition, further investigations of virulence factors are indispensable in order to elucidate *Pcal* pathogenicity and its adaptation to its different hosts, cabbage and oat. Although it is necessary to investigate the specific function of each virulence factor, our study raised the possibility that it is important to reveal the virulence mechanism of each interaction between host plants and bacteria. We believe that our continuous study will be useful for developing new strategies for sustainable disease control.

## CONCLUSION

We identified several potential virulence factors involved in *Pcal* virulence mechanism such as the T3SS, membrane transporters, transcription factors, and amino acid metabolism (Fig. 3). Moreover, predicted functions based on sequence analysis suggest that these virulence factors may be involved at different stages of the infection process. Our study highlighted the utility of transposon mutagenesis screening for identification of additional virulence factors in several bacterial pathogens. In addition, our results revealed that nearly half of genes making contributions to virulence in cabbage also make virulence contributions in oat, indicating that *Pcal* optimizes its virulence factors for pathogenicity on different host plants. Although further analysis will be needed, revealing the virulence mechanisms of *Pcal* will provide insights into a novel mechanism of host optimization.

## ACKNOWLEDGEMENTS

We thank Dr. Christina Baker for editing the manuscript. *Pcal* was kindly provided by the Nagano vegetable and ornamental crops experiment station, Nagano, Japan.

### Funding

This work was supported by JSPS KAKENHI Grant Number 19K06045 and by JST ERATO NOMURA Microbial Community Control Project, JST, Japan. The funders had no role in study design, data collection and analysis, decision to publish, or preparation of the manuscript.

### Grant Disclosures

The following grant information was disclosed by the authors:
JSPS KAKENHI: 19K06045.
JST ERATO NOMURA Microbial Community Control Project, JST, Japan.

### Competing Interests

The authors declare there are no competing interests.

### Author Contributions

- Nanami Sakata, Takako Ishiga and Yasuhiro Ishiga conceived and designed the experiments, performed the experiments, analyzed the data, contributed reagents/materials/analysis tools, prepared figures and/or tables, authored or reviewed drafts of the paper, approved the final draft.
- Haruka Saito and Viet Tru Nguyen performed the experiments, approved the final draft.

### Data Availability

The raw data is available at Figshare: Sakata, Nanami; Ishiga, Takako; Nguyen, Viet Tru; Saito, Haruka; Ishiga, Yasuhiro (2019): Transposon mutagenesis reveals *Pseudomonas cannabina* pv. *alisalensis* optimizes its virulence factors for pathogenicity on different hosts. figshare. Journal contribution. https://doi.org/10.6084/m9.figshare.9598319.v3.

### Supplemental Information

Supplemental information for this article can be found online at http://dx.doi.org/10.7717/peerj.7698#supplemental-information.

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
