# Peer review of "Transposon mutagenesis reveals Pseudomonas cannabina pv. alisalensis optimizes its virulence factors for pathogenicity on different hosts"

_PeerJ, doi:10.7717/peerj.7698_

## Round 0.1 · original submission · Major Revisions

Dear Prof. Ishiga,

Thank you for submitting your manuscript to PeerJ. You will see that the reviewers, while having generally positive comments about your paper, have concerns about your interpretation of some of your observations and your conclusions. There is also a need to edit the writing in several instances to improve clarity. Based on the number of changes that are needed, I am not able to accept your paper in its present form, however, I hope you will resubmit your paper after you have addressed each of the concerns.

I will not comment on each of the reviewers’ points but I want to offer my opinion about a few of them.

1) I agree that further substantiation of your conclusions regarding the role of amino acid biosynthesis in pathogenicity is needed.

2) I am ok with moving supplemental Figure 1 to the main text, but I leave that up to you.

3) I suggest you check with a statistician about the tests for significance in your various figures.

4) As one reviewer suggests, combining the information in Table 2 with that in Table 3 might simplify the presentation but I leave that up to you.

5) It is particularly important that you consider and respond to these concerns:
a) “ . . .could the decreased virulence result from an overall diminished fitness, independent of a specific, pathogenic interaction with the host plant?”
b) “Did any of the 60 genes identified show similarity to known effectors? Did any sequencing of insertion sites reveal evidence of upstream hrpL binding sites?”

A few additional suggestions about the writing:
Line 43: replace is with are
Line 54: replace remains with remain
Line 188: replace the with a
Line 189: replace indicating with suggesting
Line 268: The meaning of the statement is not clear (... ratio occupied by each factor?).

To expedite the review process please submit a detailed cover letter that responds to each of the points raised by the reviewers and describes the changes you have made to your manuscript to address the concerns.

Sincerely,

Gregory Martin

·

Basic reporting

The manuscript "Genome-wide transposon mutagenesis reveals Pseudomonas cannabina pv. alisalensis (Pcal )optimizes its virulence factors for pathogenicity on different hosts" by Sakata et al, reports a genetic screen to identify virulence factors in the bacterial pathogen. This work is important to understand the virulence of Pcal towards cabbage and oats. The study identified 60 Tn5 insertions that compromised virulence towards cabbage. Thirty one of those insertions also also impacted virulence on oats. The authors used the Tn5 insertion sites to identified the genes in Pcal and found that those genes belong to several functional categories and included genes known to be important in bacterial pathogenicity.
While the manuscript is easy to understand, the writing needs to be improved for clarity. Specific comments on the writing are:
Lines 29-30. A better statement for the different virulence factors needed in cabbage vs oats is: "Interestingly, almost half of the mutants that showed reduced virulence on cabbage, also showed reduced virulence on oat, …...
Line 58. Pseudomonas is not spelled correctly.
Line 61. Replace in this study, for in such study or, in that study.
Lines 73-83. It is not clear if the authors are referring to different names for the Pcal, or if they consider that Pcal is different from P. syringae pv maculicola.
Line 111. Concentration is missing information on the units.
Lines 187- 188. The authors found 60 insertions sites but only referred to 57 insertions being in coding regions, where were the other 3 insertions?
Lines 188-189. Please clarify the statement "Our results showed no virulence factors which accounted for a great ratio". What ratio are they referring to? and how is that related to the expected number of virulence factors?
Lines 209-210. The authors mentioned again 60 mutants, but from the previous section only 57 were further analyzed. It is not clear why 3 mutants appeared to be neglected and then reappear again.
Line 218-219. Does the statement "three in four" refers to the four mutants associated with amino acid biosynthesis? If that is the case, why is that assumed that those are important in colonization?
Lines 222-223. If mutations in AraC does not have a virulence defect, why are they listed as important in colonization?
Line 246. Pcal is the name of the pathogen, not the name of the disease.
Lines 251-252. Not clear how discovering those genes is evidence for the reliability of the inoculation method.
Line 279. The reference (Rico and Preston, 2008) is not relevant supporting that hypothesis.
Lines 281-282. Not sure what is the value of that statement.
Lines 306-309. It is a very strong statement that is not fully supported by the authors evidence. Not clear the link between amino acid biosynthesis and pathogenicity. Those could be detrimental for bacterial survival regardless of infection processes.
Lines 326-340. Can be simplified and focused on HexR. Superfluous information makes it hard to follow. The same can be applied for lines 345-352 and LysR.
Lines 353-361. Simplify to just mention that the study identify virulence genes that can be investigated further to dissect what is their function in the interaction between Pcal and its hosts cabbage and oats. The reference cited is not relevant.
Lines 364-365. The authors do not know that is going to be the case.

Regarding the tables and figures:
Table 1. The same name KB211 is given to two different versions of P. cannabina pv. alisalensis.
The manuscript will be improved if supplementary Figure 1 is included as a main text figure. It facilitates the understanding of the manuscript.
A picture comparing disease symptoms between wild-type bacteria and mutants is needed in the main text, especially because the authors assigned different scores of disease severity, but the pictures included in Supplementary Figures 2 and 3 showed that all the mutants are alike.
The title of Table 2 is misleading when referring to the 57 genes as "predicted to be essential" . Essential genes are required for life, which is not necessarily true for pathogenicity or virulence genes. Are the authors referring to genes required for pathogenicity?
It will be more helpful if the authors actually include the gene names and their respective functional categories in Table 2, just like they did with Table 3. That way is easier to follow the text when they are referring to specific genes (ie lines 199-206).
Methods state that mutants with reduced virulence were those with a score of less than 2 in cabbage (line 144), however, the title of table 3 indicates that reduced virulence on cabbage was scored as less or equal to 1.

Experimental design

No comment

Validity of the findings

The authors stated in the abstract that their work is insightful to understand the infection process by Pcal regarding the epiphytic phase and endophytic phase but their results do not support that. The authors should limit to state the discovery of potential virulence factors with some reference to what is known from the literature in other systems.
Lines 249- 251. The manuscript did not conducted those experiments to demonstrate that the mutated genes in Pcal are actually involved in the processes described. In lieu of the evidence, please explain which specific genes are involved in the processes described with supporting evidence from the literature. Table 2 and Figure 2 do not show that.
Lines 235. States that mutants NL37 and NN13 are not significantly different from wild-type. However, bacterial growth curve assay in Figure 1A shows that all the mutants have 2 or more logs difference in comparison with wild-type strain including NL37 and NN13. Those differences would be considered significant. It looks like there is a problem with the statistical analysis.

Reviewer 2 ·

Basic reporting

A very solid manuscript for publication with PeerJ. Definitely has potential to lead to new discoveries regarding specific host adaptation factors in a tractable broad host system.

Leaving Schrieber et al 2012 out of the introduction doesn't make sense though. This is the closest and most relevant study to the current work with the major differences being the host plants used. Please include in the intro alongside the discussion of other Tn studies in P. syringae

Table 2. I'm confused by the use of "essential" Do you mean essential for virulence? If so that wording may be far too strong for some of these genes (e.g. NL37).
I would suggest eliminating table 2 completely and porting the information for the 57 genes into table 3 with additional columns noting the actual virulence scores for both cabbage and oat.

Experimental design

How were strains verified as carrying single transposon insertions, or more specifically how were multiple insertions excluded? Were multiple genomic rescue clones sequenced to ID insertion location per strain?

The KB growth testing is not very robust and should have some statistical analysis performed (FigS4). About 1/3 of the strains do not look as though they grow similarly to WT which was reported (e.g. NN13, NL3...) I do not think this is necessarily a huge problem if growth in rich media is lower, but the differences in growth characteristics are worth noting so readers can robustly interpret the results.

Validity of the findings

The claim that cabbage requires more Pcal virulence factors than oat seems premature based on the data collected. If the Tn screen had been conducted in both cabbage and oat in parallel and fewer "vir down" strains had been found in oat overall, then maybe that claim would make more sense. There may be just as many oat-specific virulence factors that then would not show an effect in cabbage if oat had been screened first. I think it would be more valid to say that approximately half of genes making virulence contributions to disease on cabbage also make contributions to virulence in oat rather than saying oat requires fewer virulence factors.

·

Basic reporting

The quality of the writing was generally clear and straightforward. “In this study” in line 61 should be changed to “Somlyai et al inoculated…” or some analogous construction. On first reading I mistakenly thought the authors had switched to describing the conclusions of their own work.
Line 96: If the authors are going to mention the existence of “few studies reporting on in interaction… under field conditions” those few should be cited. However, the sentence could also be deleted given that the conditions being used in this study do not qualify as “field conditions” and consequently do not contribute new information to that particular gap in our knowledge
Line 99: I was also puzzled by the statement that large scale surveys have not been performed with bacteria that cause serious disease in the field. This statement is definitely not correct if the authors are indeed referring to all plant pathogenic bacteria!
Lines 249 and 364: “several genes involved in the whole infection process” infers that individual gene products are playing a role at multiple stages. It would be more appropriate to state that several genes were identified with apparent impact on virulence. “Predicted functions based on sequence analysis suggest that different members of the set may be involved at different stages of the infection process”

Experimental design

According to the manuscript, 1040 Tn5 mutants were evaluated for reduced virulence on cabbage. 60 were found to have reduced virulence on cabbage. These same 60 were then evaluated on oat and 31/60 showed reduced virulence on the monocot. The authors conclude (line 30) that more virulence factors are needed to infect cabbage than oat”. But by only testing the 60 on oat, it can’t be ruled out that there are others that are required in oat but not cabbage. The abstract (line 27) states that there are “common and specific virulence factors of both monocot and dicot plants” but the experiments were not set up to identify those specific to virulence on oats.
Additionally, given that with 1040 mutations, the study is likely evaluating ~ 20% of the total number of ORFs in the Pcal genome (assumed from genome sizes of sequenced P. syringae strains), a comprehensive statement about numbers of genes involved in virulence is not possible here. Consistent with this, the breadth and definitiveness of the title is not supported by the data presented.
The authors demonstrated that decreased virulence correlated with decreased multiplication in planta, but I did not see evidence that they had evaluated growth in culture for the virulence-impacted mutants. Not including the outright auxotrophs, could the decreased virulence result from an overall diminished fitness, independent of a specific, pathogenic interaction with the host plant?

Validity of the findings

1. The study identifies genes which, assuming unimpaired growth in culture, are involved in virulence of Pcal on cabbage.
2. There appear to be differences in genes involved in virulence on cabbage and oat, though the extent of these differences cannot be determined given the relatively limited number of mutations screened and uncertainty to which particular mutations may be generally compromising the fitness of the bacterium
3. Given the significant role of and place of Type III effectors in research on P. syringae and the evidence presented that the Type III secretion system is important for virulence, type III effectors should be addressed more directly than just lines 291-292. Did any of the 60 genes identified show similarity to known effectors? Did any sequencing of insertion sites reveal evidence of upstream hrpL binding sites? If not, this underscores the concern that 1040 mutations was too few to make a meaningful advance in our understanding of Pcal virulence factors.

---

## Round 0.2 · accepted · Accept

There are two minor points for you to address:
1) Page 3, line 23: Change tomato Latin binomial to: Solanum lycopersicum
2) page 18, line 25: correct spelling of 'virulence'

#